# High-Density Lipoprotein Cholesterol and Cognitive Function in Older Korean Adults Without Dementia: Apolipoprotein E4 as a Moderating Factor

**DOI:** 10.3390/nu17142321

**Published:** 2025-07-14

**Authors:** Young Min Choe, Hye Ji Choi, Musung Keum, Boung Chul Lee, Guk-Hee Suh, Shin Gyeom Kim, Hyun Soo Kim, Jaeuk Hwang, Dahyun Yi, Jee Wook Kim

**Affiliations:** 1Department of Neuropsychiatry, Hallym University Dongtan Sacred Heart Hospital, 7 Keunjaebong-gil, Hwaseong 18450, Gyeonggi, Republic of Korea; howlow44@daum.net (Y.M.C.); phahyethon@hanmail.net (H.J.C.); ms8989@naver.com (M.K.); suhgh@chol.com (G.-H.S.); 2Department of Psychiatry, College of Medicine, Hallym University, Chuncheon 24252, Gangwon, Republic of Korea; leeboungchul@hallym.or.kr; 3Department of Neuropsychiatry, Hallym University Hangang Sacred Heart Hospital, Seoul 07247, Republic of Korea; 4Department of Neuropsychiatry, Soonchunhyang University Bucheon Hospital, Bucheon 14584, Republic of Korea; redmensch@schmc.ac.krv; 5Department of Laboratory Medicine, Hallym University Dongtan Sacred Heart Hospital, 7 Keunjaebong-gil, Hwaseong 18450, Gyeonggi, Republic of Korea; kimhyun@hallym.or.kr; 6Department of Psychiatry, Soonchunhyang University Hospital Seoul, Seoul 04401, Republic of Korea; hju75@schmc.ac.kr; 7Institute of Human Behavioral Medicine, Medical Research Center, Seoul National University, Seoul 03080, Republic of Korea; dahyunyi@gmail.com

**Keywords:** high-density lipoprotein cholesterol, cognitive function, apolipoprotein E ε4 allele, episodic memory, Alzheimer’s disease

## Abstract

**Background:** High-density lipoprotein cholesterol (HDL-C) is known for its cardiovascular and neuroprotective effects, but its association with cognitive function remains unclear, particularly in relation to genetic factors such as apolipoprotein E ε4 (APOE4). We aimed to investigate the association between serum HDL-C levels and cognition and to examine the moderating effect of APOE4 on this relationship. **Methods:** This cross-sectional study included 196 dementia-free older adults (aged 65–90) recruited from a memory clinic and the community. Cognitive function was assessed across multiple domains using the Consortium to Establish a Registry for Alzheimer’s Disease (CERAD) battery. Serum HDL-C levels were measured, and APOE4 genotyping was performed. Multiple linear regression analyses were conducted, adjusting for age, sex, APOE4 status, education, diagnosis, vascular risk, nutritional status, physical activity, and blood biomarkers. **Results:** Higher HDL-C levels were significantly associated with better episodic memory (B = 0.109, 95% confidence interval [CI]: 0.029–0.189, *p* = 0.008) and global cognition (B = 0.130, 95% CI: 0.001–0.261, *p* = 0.049). These associations were significantly moderated by APOE4 status. In APOE4-positive individuals, HDL-C was strongly associated with both episodic memory (B = 0.357, 95% CI: 0.138–0.575, *p* = 0.003) and global cognition (B = 0.519, 95% CI: 0.220–0.818, *p* = 0.002), but no such associations were observed in APOE4-negative participants. **Conclusions:** This study indicates a significant association between serum HDL-C levels and cognitive function, particularly in episodic memory and global cognition, with APOE4 status potentially moderating this relationship. While these findings may suggest a protective role of HDL-C in individuals at increased genetic risk due to APOE4, they should be interpreted with caution given the cross-sectional design. Future longitudinal and mechanistic studies are warranted to clarify causality and potential clinical implications.

## 1. Introduction

High-density lipoprotein cholesterol (HDL-C) is well recognized for its cardiovascular benefits and has recently garnered attention for its potential neuroprotective effects [1,2]. This lipoprotein contributes to cognitive resilience by reducing neuroinflammation, mitigating oxidative stress, and supporting cholesterol metabolism in the brain [3,4]. However, studies investigating the relationship between HDL-C and cognitive function or dementia have produced inconsistent findings [5,6,7,8,9,10,11,12].

These inconsistencies may arise from heterogeneity in study designs, variations in cognitive test domains, genetic factors such as the apolipoprotein E (APOE) genotype, and demographic differences in age and sex. Some research suggests that HDL-C may have domain-specific effects, with its influence varying across cognitive domains [12,13,14]. While higher HDL-C levels have been associated with better memory performance in some studies [13], another study has reported stronger associations with attention and executive functions [14]. Additionally, a separate study identified sex-specific associations of HDL-C with multiple cognitive domains, including attention, executive functions, and memory, further underscoring the complexity of these relationships [12].

The apolipoprotein E ε4 allele (APOE4), a major genetic risk factor for Alzheimer’s disease (AD) and cognitive decline, has been implicated in impairments in lipid transport, amyloid-beta clearance, and neurovascular integrity [15,16,17]. These mechanisms may attenuate HDL-C’s neuroprotective effects, such as promoting cholesterol efflux and reducing neuroinflammation. Although APOE4 is widely studied as a risk factor, relatively few studies have explicitly tested its moderating effect on the relationship between HDL-C and cognitive function. One study reported interactions between HDL-C and APOE genotype limited to memory [14], but broader moderation effects across multiple cognitive domains remain poorly understood. Understanding the interaction between HDL-C and APOE4 has important clinical implications. If HDL-C modulates APOE4-related cognitive vulnerability, it may support early risk stratification and guide lifestyle or lipid-targeted interventions in genetically at-risk populations.

This study aims to examine the association between serum HDL-C levels and cognitive function across specific domains—including episodic memory, executive function, and global cognition—and determine whether this relationship is moderated by APOE4 status in older adults without dementia.

## 2. Materials and Methods

### 2.1. Participants

This study is a cross-sectional analysis of baseline data from the General Lifestyle and Alzheimer’s Disease (GLAD) study, an ongoing prospective cohort that began in 2020. As of July 2022, a total of 196 community-dwelling older adults aged 65–90 years who had not been clinically diagnosed with dementia were enrolled. Of these, 83 participants were classified as cognitively normal (CN), and 113 were identified as having mild cognitive impairment (MCI).

Participants were recruited through two complementary approaches. First, individuals who attended a dementia screening program at the memory clinic of Hallym University Dongtan Sacred Heart Hospital in Hwaseong, South Korea were invited for eligibility assessment. Second, additional volunteers were recruited from the local community through referrals by existing participants, family members, or acquaintances. These community-based participants were selected to ensure diversity in demographic characteristics and cognitive status, with the goal of improving the epidemiological representativeness of the study sample among older Korean adults.

The CN group consisted of participants with a Clinical Dementia Rating [18] score of 0 and no diagnosis of MCI or dementia. All participants with MCI met the current consensus criteria for amnestic MCI, including memory complaints confirmed by an informant, objective memory impairment, preservation of global cognitive function, independence in functional activities, and the absence of dementia. In the objective memory impairment assessment, the age-, education-, and sex-adjusted z-score was <−1.0 on at least one of four episodic memory subtests—word list memory, word list recall, word list recognition, and the constructional recall test)—from the Korean version of the Consortium to Establish a Registry for Alzheimer’s Disease (CERAD) neuropsychological battery [19,20,21]. All MCI individuals had a Clinical Dementia Rating score of 0.5. The exclusion criteria were the presence of a major psychiatric illness, a significant neurological or medical condition, or a comorbidity that could affect mental functioning; illiteracy; the presence of visual/hearing difficulties and/or severe communication or behavioral problems that would make clinical examinations difficult; and the use of an investigational drug.

The study protocol was approved by the Institutional Review Board of the Hallym University Dongtan Sacred Heart Hospital and was conducted it in accordance with the recommendations of the current version of the Declaration of Helsinki. The participants or their legal representatives provided informed consent.

### 2.2. Clinical Assessments

All participants underwent standardized clinical assessments by trained psychiatrists based on the GLAD study’s clinical assessment protocol, which incorporates the CERAD clinical and neuropsychological battery [19,20]. The CERAD neuropsychological battery [21] was administered by licensed psychologists who had experience working with older adults. All clinical decisions and assessments were based on the consensus of a panel of psychiatrists and psychologists with expertise in dementia.

Cognitive outcomes were assessed across three domains: episodic memory, non-memory, and global cognition. Episodic memory was evaluated using four tests from the CERAD battery—word list memory, word list recall, word list recognition, and constructional recall—and their summed score was termed the episodic memory score (EMS). Episodic memory decline represents an early cognitive marker of AD [22,23,24,25,26,27]. Non-memory cognition was assessed using three tests—verbal fluency (executive function/attention/language), the modified Boston Naming Test (language), and constructional praxis (visuospatial function)—with the combined score referred to as the non-memory score (NMS) [28]. Global cognition was measured using the CERAD total score (TS), calculated by summing the scores of all seven tests in the battery [29]. To standardize interpretation across the manuscript, these cognitive scores—EMS, NMS, and TS—are consistently used to represent episodic memory, non-memory, and global cognitive function, respectively.

Vascular risks were assessed based on data collected by trained researchers during systematic interviews of the participants and their family members. The vascular risk score (VRS) was calculated based on the number of vascular risks, including hypertension, diabetes mellitus, dyslipidemia, coronary heart disease, transient ischemic attack, and stroke, present and was reported as a percentage [30]. The Body Mass Index (BMI) was determined using the individual’s weight in kilograms divided by their height in meters squared. Physical activities were evaluated using the Korean version of the Physical Activity Scale for the Elderly (PASE) [31,32], which has been tested for its reliability and validity. Trained researchers evaluated the participants’ frequency, duration, and intensity of activities performed during leisure, household, and occupational activities during the previous week. A weighted total physical activity score based on PASE subscores reflecting these activities was calculated.

Nutritional status was assessed using the Mini Nutritional Assessment (MNA) tool [33], a validated instrument designed for use in older populations. The MNA comprises two components: a screening section (maximum score of 14) and an assessment section (maximum score of 16). The sum of these two sections yields the MNA total score, which ranges from 0 to 30. The MNA total score was subsequently included to assess the potential confounding effect of overall nutritional status on the relationship between HDL-C and cognitive outcomes. The MNA includes evaluation of any change in food intake over the past 3 months due to loss of appetite, digestive problems, or chewing/swallowing difficulties—and dietary patterns, including intake of protein and fruits or vegetables. Protein intake levels were categorized as ‘low’, ‘medium’, or ‘high’ based on adherence to three key consumption markers: daily dairy, weekly legumes or eggs (two or more servings), and daily meat, fish, or poultry; intake was classified as low (0–1 marker met), medium (2 markers met), and high (all 3 markers met). Fruit and vegetable intake was classified as ‘high’ if participants reported consuming two or more servings per day, and ‘low’ otherwise. To acquire accurate information for all assessments, reliable informants were also interviewed.

### 2.3. Measuring Serum Levels of HDL-C and Other Blood Biomarkers

Morning (8–9 A.M.) blood samples were collected through venipuncture. HDL-C and low-density lipoprotein-cholesterol (LDL-C), albumin, and glucose were also measured using a COBAS c702 analyzer and dedicated reagents (Roche Diagnostics, Mannheim, Germany).

### 2.4. APOE4 Genotyping

Blood samples were collected in EDTA anticoagulated vacutainer tube. Genomic DNA was extracted using QIAamp DSP DNA Blood mini kit (QIAGEN, Hilden, Germany) and QIAcube HT System (QIAGEN, Hilden, Germany). The APOE genotyping was performed using a Seeplex ApoE ACE Genotyping Kit (Seegene, Seoul, Republic of Korea) and ProFlex PCR system (ThermoFisher Scientific, Waltham, MA, USA) according to the manufacturer’s instruction. The PCR product was analyzed using a capillary electrophoresis device (QIAxcel Advanced System, QIAGEN, Hilden, Germany), and interpreted as ε2/ε2, ε2/ε3, ε2/ε4, ε3/ε3, ε3/ε4, or ε4/ε4 according to the electrophoresis pattern and manufacturer’s instruction. Participants were defined as APOE4-positive if they had at least one ε4 allele.

### 2.5. Statistical Analysis

To examine the relationships between HDL-C levels and cognitive function, multiple linear regression analyses were performed using HDL-C as the independent variable—treated either as a continuous (in mg/dL) or a categorical variable depending on the specific analysis—and cognitive scores (EMS, NMS, and TS) as dependent variables. In the primary analyses, HDL-C was modeled as a continuous variable to assess linear associations. For additional analyses, HDL-C was categorized into three groups: ‘low’ (<50 mg/dL), based on the Mayo Clinic criteria (https://www.mayoclinic.org/diseases-conditions/high-blood-cholesterol/in-depth/hdl-cholesterol/art-20046388 (accessed on 5 May 2025)), which is particularly relevant given the predominantly female study population (over 70%). Among those with HDL-C ≥ 50 mg/dL, two additional strata were defined using the median: ‘medium’ (50–60 mg/dL) and ‘high’ (>60 mg/dL), to ensure a balanced distribution of participants across the categories.

To examine the independent association between HDL-C levels and cognitive unction, multiple linear regression models were adjusted for a range of covariates that are known to affect both HDL-C and cognitive function. These included age, sex, APOE4 status, education, clinical diagnosis (CN or MCI), vascular risk factors, BMI, physical activity, dietary patterns (e.g., protein and fruit/vegetable intake), and blood nutritional markers such as albumin, glucose, and LDL-C levels [13,34,35,36,37,38,39,40]. Two models were used, each controlling for the covariates in a stepwise manner. The first model included age, sex, APOE4 status, education, clinical diagnosis, as covariates; and the second model included those covariates plus VRS, BMI, PASE total score, protein intake, fruit or vegetables intake, albumin, fasting glucose, and LDL-C levels. To confirm the robustness of our regression analyses, we assessed the assumptions of normality and homoscedasticity of residuals and verified the absence of collinearity by utilizing normal probability plots, scatter plots, and variance inflation factor (VIF) values.

The moderating effect of APOE4-positivity on the association between HDL-C levels and cognitive function was examined using multiple linear regression models that included HDL-C, APOE4 status, and their interaction term (HDL-C × APOE4). These models also adjusted for all other previously described covariates, excluding APOE4 as a covariate. When a significant interaction was observed, stratified analyses were conducted separately within APOE4-positive and APOE4-negative subgroups to further explore group-specific associations.

The same sensitivity analyses were also performed in individuals with no decrease in food intake over the past 3 months for any reason to eliminate any influence of a physical or mental condition potentially related to the HDL-C levels and cognitive function status. All statistical analyses were performed using SPSS Statistics software ver. 28 (IBM, Armonk, NY, USA), with *p* < 0.05 were considered statistically significant. Given the assessment of multiple cognitive outcomes, a Bonferroni correction was applied (adjusted significance threshold: *p* < 0.0125 based on four comparisons), and relevant findings are noted in the Section 4.

## 3. Results

### 3.1. Participants

Table 1 outlines the demographic and clinical characteristics of the study cohort, comprising 156 APOE4-negative and 40 APOE4-positive participants. There were no statistically significant differences in the demographic variables or other clinical characteristics between these two groups.

### 3.2. Association of the Serum HDL-C Levels with Cognitive Function

Serum HDL-C levels were positively associated with TS (B = 0.130, 95% confidence interval [CI]: 0.001–0.261, *p* = 0.049) and EMS (B = 0.109, 95% CI: 0.029–0.189, *p* = 0.008) after adjusting all covariates (Model 2). In contrast, no significant associations were found for verbal fluency, the modified Boston Naming Test, or constructional praxis (Table 2; Figure 1A,B).

### 3.3. APOE4 Moderation of the Association Between the Serum HDL-C Levels and Cognitive Function

The interaction between serum HDL-C levels and APOE4 positivity significantly influenced TS (B = 0.379, 95% CI: 0.082–0.675, *p* = 0.013) and EMS (B = 0.236, 95% CI: 0.057–0.416, *p* = 0.010), but had no significant effect on verbal fluency, the modified Boston Naming Test, or constructional praxis. These findings indicate that APOE4 status modulates the relationship between serum HDL-C levels and both episodic memory and global cognition, while showing no moderating effects on non-memory cognitive tests (Table 3). In addition to the interaction effects, APOE4-positivity was independently associated with significantly lower TS (B = −22.733, 95% CI: −39.451–−6.015, *p* = 0.008) and EMS (B = −15.813, 95% CI: −25.820–−5.806, *p* = 0.002). These findings reaffirm the detrimental impact of APOE4 on episodic and global cognitive function, independent of serum HDL-C levels.

### 3.4. Subgroup Analyses Based on APOE4 Status

The analysis stratified by APOE4 status revealed significant associations between serum HDL-C levels and both TS (B = 0.519, 95% CI: 0.220–0.818, *p* = 0.002) and EMS (B = 0.357, 95% CI: 0.138–0.575, *p* = 0.003) in the APOE4-positive group after adjusting all covariates (models 2), whereas no such associations were identified in the APOE4-negative group. These results are shown in Table 4 and Figure 1C–F, which provide complementary statistical and visual representations.

### 3.5. Association of the Serum HDL-C Level Groups with Cognitive Function

In the categorical analysis of HDL-C levels, participants in the high HDL-C group had significantly higher EMS compared to those in the low HDL-C group (B = 3.239, 95% CI: 0.766–5.713, *p* = 0.011) after adjusting all covariates (models 2). Although a positive trend was observed for total cognitive scores (TS) in the high HDL-C group, the association did not reach statistical significance (B = 3.982, 95% CI: −0.214–8.178, *p* = 0.063) (Table 5). Stratified analyses by APOE4 status revealed that among APOE4-positive individuals, those with high HDL-C levels showed significantly higher TS (B = 13.945, 95% CI: 3.754–24.137, *p* = 0.010) and EMS (B = 10.677, 95% CI: 3.632–17.722, *p* = 0.005) after adjusting all covariates (models 2), whereas no significant associations were observed in the APOE4-negative group (Table 5).

### 3.6. Sensitivity Analyses

The sensitivity analysis conducted on participants who had not experienced a decrease in food intake over the past 3 months yielded similar results for both TS and EMS (Table 6).

## 4. Discussion

This study explored the relationship between serum HDL-C levels and cognitive function across various domains, with a specific focus on EMS, NMS, and TS. Additionally, it examined the moderating effects of APOE4 on these associations. Key findings revealed a positive association between higher HDL-C levels and EMS, as well as TS, but not NMS. APOE4 status significantly moderated these relationships, with stronger associations observed in APOE4-positive participants. Sensitivity analyses excluding participants with decreased food intake over the past three months supported the robustness of these findings.

Our findings align with prior research on HDL-C’s domain-specific effects on cognitive function. For instance, several studies have reported positive associations between HDL-C levels and memory performance [13,37], consistent with the current study’s EMS results. Conversely, one study has demonstrated stronger associations with attention and executive functions [14]. Additionally, a separate study highlighted sex-specific associations with multiple cognitive domains, including attention, executive functions, and memory [12]. These variations suggest that HDL-C’s cognitive benefits may differ across populations, sexes, and cognitive domains. Furthermore, earlier meta-analyses have noted inconsistent findings, often attributed to methodological heterogeneity and confounding factors [5]. To address these limitations, the present study controlled for variables such as LDL-C, vascular risks, and nutritional factors, aiming to reduce potential biases and provide clearer insights into HDL-C’s cognitive implications.

The study’s finding that APOE4 moderates the relationship between HDL-C levels and cognitive function underscores the complexity of this relationship. Among APOE4-positive individuals, those with higher HDL-C levels demonstrated better performance in EMS and TS, suggesting that HDL-C may be associated with a reduced impact of APOE4’s detrimental effects on lipid transport, amyloid-beta clearance, and neurovascular health [16,17]. Importantly, the strongest effects were observed in episodic memory, which aligns with the earliest cognitive decline associated with AD [22,23,24,25,26,27]. This study builds on previous research, which noted domain-specific HDL-C effects in APOE genotypes [14], and highlights the clinical importance of episodic memory as a potential early marker for AD. In contrast to these significant associations observed for episodic memory and global cognition, no associations were found between HDL-C levels and other cognitive domains such as verbal fluency, naming, and constructional praxis. These null findings highlight the domain-specific nature of HDL-C’s effects and suggest that its potential cognitive benefits may be limited to memory-related processes. Further research is needed to clarify whether this selectivity reflects underlying neurobiological mechanisms or sample-specific characteristics.

However, conflicting findings have also been reported. For instance, a study focusing on middle-aged individuals did not observe a significant moderating effect of APOE4 [13]. One possible explanation is that the shorter follow-up period in this study may not have allowed sufficient time for APOE4’s moderating effects to manifest. Additionally, the absence of APOE4’s moderating effects in middle-aged populations might be due to differences in neuropathological processes between middle age and older adulthood. During middle age, compensatory mechanisms such as greater neuroplasticity or the availability of alternative pathways for lipid metabolism and amyloid-beta clearance may mitigate APOE4’s negative impact [41,42]. Conversely, these mechanisms might weaken in later life, allowing APOE4-related vulnerabilities to emerge more clearly. Differences in vascular health, which is often better in middle age, could also contribute to the lack of observable effects in younger populations [43].

Beyond these interaction effects, APOE4-positivity itself was strongly associated with lower global and episodic memory scores, reinforcing its established role in cognitive vulnerability. The significant main effects observed in both TS and EMS suggest that APOE4 may contribute to broad cognitive decline regardless of HDL-C levels, and further highlight the relevance of stratifying by APOE genotype in cognitive aging research.

Evidence suggests that HDL-C is involved in various neuroprotective processes that may contribute to cognitive resilience, particularly in individuals at increased risk for AD. Among the proposed mechanisms, the most compelling human-based evidence supports HDL-C’s role in promoting amyloid-beta clearance and preserving vascular integrity [16,17]. In this context, HDL-C enhances lipid transport and facilitates amyloid-beta efflux through pathways involving apolipoprotein E and the ABCA1/ABCG1 transporters [16,44,45], which is especially relevant in APOE4 carriers with impaired lipid metabolism and reduced amyloid clearance efficiency [16]. In addition, HDL-C appears to exert vasoprotective effects—such as maintaining endothelial function and reducing neurovascular inflammation—that may help preserve blood–brain barrier integrity and cerebral perfusion [17,46]. These vascular benefits may be particularly important in APOE4 carriers, who are more vulnerable to neurovascular dysfunction. Although additional mechanisms—such as antioxidant activity via paraoxonase 1 (PON1) [47,48], signaling modulation through sphingosine-1-phosphate (S1P) [49], and activation of the PI3K/Akt pathway to promote neuronal survival and synaptic plasticity [50]—have been proposed, current human evidence remains limited. In addition, HDL-C may regulate neuroinflammation through astrocytic and microglial pathways [6,51], and influence brain-derived neurotrophic factor (BDNF) expression, which supports hippocampal integrity and memory [52]. Given the reduced BDNF levels observed in APOE4 carriers, this neurotrophic mechanism may represent an additional route through which HDL-C mitigates synaptic vulnerability. Future studies should explicitly test these pathways using longitudinal and biomarker-based approaches. Overall, our findings support the hypothesis that HDL-C may reduce APOE4-related AD risk via mechanisms involving lipid metabolism, amyloid clearance, vascular protection, and neurotrophic and anti-inflammatory support.

Although our findings suggest a beneficial association between HDL-C and cognitive function, evidence from longitudinal and interventional studies remains inconclusive. Prospective cohort studies have reported that higher HDL-C levels are associated with slower cognitive decline and a reduced risk of dementia [10]. In contrast, Mendelian randomization studies have shown inconsistent results, with some reporting no causal effect of genetically elevated HDL-C on AD risk [53,54]. Similarly, randomized controlled trials aiming to raise HDL-C levels—particularly through cholesteryl ester transfer protein inhibitors—have not demonstrated clear cognitive benefits [55]. These discrepancies underscore the importance of HDL functionality, including cholesterol efflux capacity, anti-inflammatory effects, and particle composition, rather than HDL-C concentration alone. Therefore, while our findings support the hypothesis that HDL-C may be associated with APOE4-related cognitive vulnerability, they should be interpreted with caution and validated through future mechanistic and longitudinal studies.

Dietary patterns such as the Mediterranean diet may enhance HDL functionality and support cognitive health. Rich in unsaturated fats and polyphenols, this diet has been shown to improve HDL particle composition, cholesterol efflux, and antioxidant capacity [56,57,58,59,60]. These effects may help explain its association with slower cognitive decline and reduced AD risk [61,62]. Benefits may be especially relevant for APOE4 carriers, who are prone to lipid dysregulation and neuroinflammation. Nutrition-based strategies targeting HDL function may thus offer a promising approach to cognitive prevention in at-risk individuals. Future research should integrate HDL function, diet, and genetic risk to identify modifiable pathways in cognitive aging.

This study has several strengths, including its robust methodology, comprehensive adjustments for confounders, and sensitivity analyses to account for potential biases from dietary decline. The use of standardized cognitive assessments and stratification by APOE4 status allowed for nuanced insights. To the best of our knowledge, this study is the first to show a moderating effect of APOE4 on the association between the serum HDL-C levels and episodic memory or global cognition in humans.

This study has several limitations that should be considered. First, as a cross-sectional study, it cannot establish causal relationships between HDL-C levels and cognitive function. Longitudinal studies are needed to clarify the temporal dynamics of this association. Second, the study did not measure AD-related pathological biomarkers, such as amyloid-beta and tau proteins, which could provide deeper insights into the relationship between HDL-C levels and AD pathogenesis. Third, it remains unclear whether APOE4 acts primarily as a moderator or as a principal driver in the observed association between HDL-C levels and cognitive function. However, the positive association between HDL-C and episodic memory in APOE4-positive participants is a notable finding. Fourth, previous studies have suggested that extremely high HDL-C levels (>80 mg/dL) may negatively affect cognitive function, but this was not explored in detail here due to the small sample size of participants with very high HDL-C (*n* = 3). Fifth, HDL-C was categorized based on the Mayo Clinic criteria for women, which may not be directly generalizable to Korean men or to the broader Korean population. These thresholds were chosen to ensure a balanced distribution of participants across groups, particularly considering the predominantly female sample. Nevertheless, our primary analyses treated HDL-C as a continuous variable and yielded consistent associations with cognitive outcomes, supporting the robustness of our findings regardless of the categorization scheme. Sixth, participants were recruited from a memory clinic and through community outreach, which may have introduced selection bias and limited the generalizability of the findings. The sample may include individuals with greater cognitive concerns or higher health awareness compared to the broader older adult population. For instance, a gender imbalance was observed, with a higher proportion of female participants compared to male participants. Therefore, future studies using population-based samples are warranted to confirm the robustness and applicability of these findings. Seventh, although multiple cognitive outcomes were assessed, no formal correction for multiple comparisons was applied due to the exploratory nature of the study. Notably, the association between HDL-C and EMS remained statistically significant even after Bonferroni adjustment (*p* < 0.0125), based on four cognitive outcomes. Nevertheless, the findings should be interpreted with caution given the potential risk of type I error. Eighth, although significant associations were observed in the APOE4-positive subgroup, the relatively small sample size (*n* = 40) may limit the stability and generalizability of the estimates. The wide confidence intervals around some of the beta coefficients reflect statistical uncertainty, and the results should therefore be interpreted with caution. Nevertheless, the consistency of findings across both Model 1 and Model 2—before and after adjusting for vascular, metabolic, and nutritional covariates—provides some support for the robustness of the observed associations. Finally, while dietary and nutritional factors could influence HDL-C levels, the sensitivity analysis excluding participants with dietary decline and the inclusion of dietary biomarkers and patterns as covariates produced similar results, reducing the likelihood of these factors confounding the findings.

To further validate this, we conducted additional regression analyses incorporating the total MNA score as a comprehensive indicator of nutritional status. The associations between HDL-C levels and both episodic memory and global cognition remained robust—particularly among APOE4-positive individuals—even after adjusting for the MNA score (Appendix A). These findings support the independent relevance of HDL-C to cognitive outcomes beyond general nutritional condition.

Future research should focus on longitudinal studies to establish causality and explore temporal relationships between HDL-C levels and cognitive trajectories. Investigating the impact of interventions aimed at modulating HDL-C, such as lifestyle or pharmacological approaches, in diverse populations and across APOE genotypes could yield valuable insights. Furthermore, examining potential thresholds for HDL-C levels, including the effects of very high HDL-C, is crucial for refining clinical recommendations.

## 5. Conclusions

This study indicates a significant association between serum HDL-C levels and cognitive function, particularly in episodic memory and global cognition, with APOE4 status potentially moderating this relationship. While these findings may suggest a protective role of HDL-C in individuals at increased genetic risk due to APOE4, they should be interpreted with caution given the cross-sectional design. Future longitudinal and mechanistic studies are warranted to clarify causality and potential clinical implications.

## Figures and Tables

**Figure 1 nutrients-17-02321-f001:**
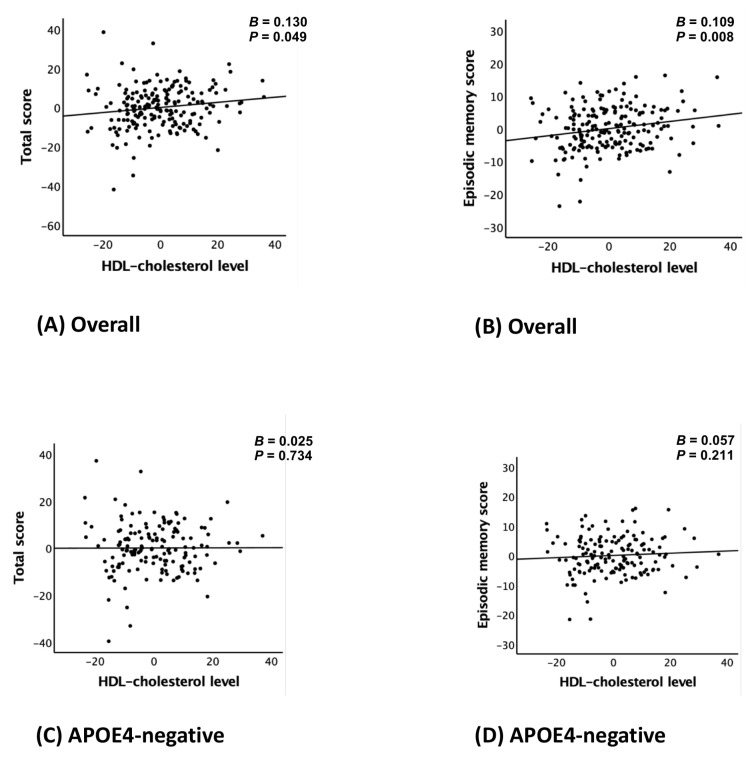
Partial regression plot for the association between the HDL-cholesterol level and CERAD total score (**A**,**C**,**E**) or episodic memory score (**B**,**D**,**F**) in non-demented older adults: (**A**,**B**) overall, (**C**,**D**) APOE4-negative, and (**E**,**F**) APOE4-positive group. Abbreviations: high-density lipoprotein (HDL); CERAD, the Consortium to Establish a Registry for Alzheimer’s Disease; APOE4, apolipoprotein ε4 allele. Footnotes: error bars indicate standard errors. Multiple linear regression analyses were performed after adjusting for all confounders.

**Table 1 nutrients-17-02321-t001:** Demographic and clinical characteristics of the participants according to APOE4 status.

	Overall	APOE4-Negative	APOE4-Positive	*p*
*n*	196	156	40	
Age, y	72.65 (5.95)	72.95 (5.96)	71.50 (5.86)	0.170 ^a^
Female, *n* (%)	138 (70.41)	106 (67.95)	32 (80.00)	0.136 ^b^
Education, y	9.62 (4.51)	9.61 (4.55)	9.68 (4.38)	0.934 ^a^
MCI, *n* (%)	113 (57.65)	88 (56.41)	25 (62.50)	0.487 ^b^
VRS, %	23.98 (18.58)	23.93 (19.14)	24.17 (16.43)	0.943 ^a^
MMSE	25.58 (3.45)	25.52 (3.46)	25.83 (3.43)	0.618 ^a^
Dietary pattern including food types				
Protein, *n* (%)				0.410 ^b^
high	27 (13.78)	19 (12.18)	8 (20.00)	
medium	74 (37.76)	59 (37.82)	15 (37.50)	
low	95 (48.47)	78 (50.00)	17 (42.50)	
Fruit or vegetables, *n* (%)				0.795 ^b^
high	119 (60.71)	62 (39.74)	15 (37.50)	
low	77 (39.29)	94 (60.26)	25 (62.50)	
Decrease in food intake over the past three months				1.00 ^c^
no, *n* (%)	182 (92.86)	145 (92.95)	37 (92.50)	
yes, *n* (%)	14 (7.14)	11 (7.05)	3 (7.50)	
Serum nutritional markers				
Albumin, g/dL	4.57 (0.26)	4.57 (0.26)	4.60 (0.25)	0.465 ^a^
Glucose, fasting, mg/dL	108.15 (19.94)	108.46 (21.02)	106.87 (14.87)	0.660 ^a^
HDL-cholesterol, mg/dL	54.64 (12.96)	54.51 (12.89)	55.21 (13.38)	0.765 ^a^
HDL-cholesterol, *n* (%)				0.481 ^b^
high	60 (30.61)	48 (30.77)	12 (30.00)	
medium	58 (29.59)	44 (28.21)	14 (35.00)	
low	76 (38.78)	64 (41.03)	12 (30.00)	
LDL-cholesterol, mg/dL	96.41 (33.82)	96.10 (35.42)	97.68 (26.64)	0.796 ^a^
BMI, kg/m^2^	24.82 (3.41)	24.78 (3.41)	25.03 (3.40)	0.680 ^a^
Mini nutrition assessment				
Screening score	12.60 (2.08)	12.54 (2.13)	12.83 (1.91)	0.439 ^a^
Assessment score	13.01 (1.75)	12.93 (1.78)	13.33 (1.61)	0.200 ^a^
Total score	25.60 (3.33)	25.46 (3.43)	26.15 (2.89)	0.247 ^a^
Physical activity				
PASE total score	64.77 (46.21)	64.45 (47.19)	66.04 (42.70)	0.847 ^a^
Multi-domains of Cognition				
Memory score				
EMS	35.10 (9.48)	35.17 (9.47)	34.83 (9.67)	0.840 ^a^
Non-memory score				
Verbal fluency (for executive function/attention)	12.79 (4.00)	12.75 (4.19)	12.93 (3.19)	0.806 ^a^
Modified Boston naming test (for language)	11.53 (2.52)	11.45 (2.54)	11.85 (2.46)	0.371 ^a^
Constructional praxis (for visual spatial skill)	9.92 (1.80)	9.85 (1.89)	10.23 (1.37)	0.237 ^a^
Global cognition				
TS	69.98 (15.61)	70.00 (16.15)	69.90 (13.52)	0.971 ^a^

Abbreviations: *p*, *p*-value; APOE4, apolipoprotein E ε4 allele; MCI, mild cognitive impairment; VRS vascular risk score; MMSE, Mini-Mental State Examination; BMI, body mass index; PASE, physical activity scale for the elderly; EMS, episodic memory score; total score of the Consortium to Establish a Registry for Alzheimer’s Disease. Data are expressed as mean (standard deviation), unless otherwise indicated. ^a^ by student t test ^b^ by chi-square test. ^c^ by fisher exact test.

**Table 2 nutrients-17-02321-t002:** Results of the multiple linear regression analyses of the association between the HDL-cholesterol level and cognitive decline.

	B	95% CI	β	*p*
TS				
Model 1	0.143	0.015–0.271	0.118	0.028
Model 2	0.130	0.001–0.261	0.108	0.049
EMS				
Model 1	0.120	0.041–0.198	0.163	0.003
Model 2	0.109	0.029–0.189	0.149	0.008
NMS				
Verbal fluency				
Model 1	0.031	−0.008–0.070	0.100	0.118
Model 2	0.034	−0.006–0.075	0.111	0.095
Boston naming test				
Model 1	−0.001	−0.027–0.025	−0.004	0.950
Model 2	−0.001	−0.028–0.026	−0.007	0.920
Constructional praxis				
Model 1	0.019	0.001–0.036	0.133	0.039
Model 2	0.018	0.001–0.037	0.131	0.053

Abbreviations: B, regression coefficient, CI, confidence interval; β, standardized beta; *p*, *p*-value; APOE4, apolipoprotein E ε4 allele; EMS, episodic memory score; NMS, non-memory score; TS, total score of the Consortium to Establish a Registry for Alzheimer’s Disease; VRS, vascular risk score, BMI, body mass index; PASE, physical activity scale for the elderly. The first model included age, sex, APOE4, education, and clinical diagnosis as covariates; the second model included those covariates plus VRS, BMI, PASE total score, protein intake, fruit/vegetable, albumin, fasting glucose, and LDL-cholesterol.

**Table 3 nutrients-17-02321-t003:** Results of multiple linear regression analyses that included interaction terms for the association between HDL-cholesterol and APOE4-positivity in predicting cognitive decline.

	B	95% CI	β	*p*
TS				
HDL-cholesterol level	0.063	−0.077–0.204	0.052	0.375
APOE4-positivity	−22.733	−39.451–−6.015	−0.571	0.008
HDL-cholesterol level × APOE4-positivity	0.379	0.082–0.675	0.540	0.013
EMS				
HDL-cholesterol level	0.051	−0.032–0.134	0.072	0.228
APOE4-positivity	−15.813	−25.820–−5.806	−0.677	0.002
HDL-cholesterol × APOE4-positivity	0.236	0.057–0.416	0.570	0.010
NMS				
Verbal fluency				
HDL-cholesterol level	0.017	−0.026–0.060	0.054	0.445
APOE4-positivity	−3.404	−8.541–1.733	−0.334	0.193
HDL-cholesterol × APOE4-positivity	0.067	−0.024–0.158	0.376	0.148
Boston naming test				
HDL-cholesterol level	−0.009	−0.037–0.020	−0.044	0.557
APOE4-positivity	−1.770	−5.190–1.650	−0.275	0.309
HDL-cholesterol × APOE4-positivity	0.037	−0.024–0.098	0.326	0.233
Constructional praxis				
HDL-cholesterol level	0.016	−0.003–0.036	0.117	0.096
APOE4-positivity	−1.608	−3.909–0.694	−0.349	0.170
HDL-cholesterol × APOE4-positivity	0.033	−0.008–0.074	0.408	0.112

Abbreviations: B, regression coefficient, CI, confidence interval; β, standardized beta; *p*, *p*-value; APOE4, apolipoprotein ε4 allele; TS, total score of the Consortium to Establish a Registry for Alzheimer’s Disease; EMS, episodic memory score; NMS, non-memory score. To explore the moderating effects of APOE4-positivity on the associations between HDL-cholesterol level and cognition, i.e., TS, EMS, and NMS, the multiple linear regression analyses were performed including two-way interaction terms between HDL-cholesterol level and cognition as additional independent variables.

**Table 4 nutrients-17-02321-t004:** Results of the multiple linear regression analyses of the association between the HDL-cholesterol level and cognitive decline according to APOE4 subgroup.

	B	95% CI	β	*p*
TS				
APOE4-negative				
Model 1	0.034	−0.1120–0.180	0.027	0.648
Model 2	0.025	−0.0120–0.170	0.020	0.734
APOE4-positive				
Model 1	0.577	0.376–0.778	0.640	<0.001
Model 2	0.519	0.220–0.818	0.575	0.002
EMS				
APOE4-negative				
Model 1	0.064	−0.025–0.154	0.088	0.156
Model 2	0.057	−0.033–0.148	0.078	0.211
APOE4-positive				
Model 1	0.385	0.232–0.537	0.597	<0.001
Model 2	0.357	0.138–0.575	0.554	0.003

Abbreviations: B, regression coefficient, CI, confidence interval; β, standardized beta; *p*, *p*-value; APOE4, apolipoprotein E ε4 allele; EMS, episodic memory score; TS, total score of the Consortium to Establish a Registry for Alzheimer’s Disease; VRS, vascular risk score, BMI, body mass index; PASE, physical activity scale for the elderly. The first model included age, sex, APOE4, education, and clinical diagnosis as covariates; the second model included those covariates plus VRS, BMI, PASE total score, protein intake, fruit/vegetable, albumin, fasting glucose, and LDL-cholesterol.

**Table 5 nutrients-17-02321-t005:** Results of the multiple linear regression analyses of the association between the stratified HDL-cholesterol level and cognitive decline according to APOE4 subgroup.

	B	95% CI	β	*p*
TS				
Overall				
Model 1				
High HDL-cholesterol	4.671	0.244–9.097	0.138	0.039
Medium HDL-cholesterol	2.710	−1.759–7.178	0.079	0.233
Low HDL-cholesterol	Reference			
Model 2				
High HDL-cholesterol	3.669	−0.215–7.552	0.108	0.064
Medium HDL-cholesterol	2.347	−1.516–6.211	0.069	0.232
Low HDL-cholesterol	Reference			
APOE4-negative				
Model 1				
High HDL-cholesterol	1.141	−3.276–5.557	0.033	0.611
Medium HDL-cholesterol	0.250	−4.116–4.616	0.007	0.910
Low HDL-cholesterol	Reference			
Model 2				
High HDL-cholesterol	1.019	−3.414–5.452	0.029	0.650
Medium HDL-cholesterol	−0.461	−4.796–3.875	−0.013	0.834
Low HDL-cholesterol	Reference			
APOE4-positive				
Model 1				
High HDL-cholesterol	16.235	8.473–23.997	0.623	<0.001
Medium HDL-cholesterol	12.061	4.409–19.712	0.472	0.003
Low HDL-cholesterol	Reference			
Model 2				
High HDL-cholesterol	13.945	3.754–24.137	0.535	0.010
Medium HDL-cholesterol	10.866	1.692–20.039	0.425	0.022
Low HDL-cholesterol	Reference			
EMS				
Overall				
Model 1				
High HDL-cholesterol	3.555	1.131–5.978	0.174	0.004
Medium HDL-cholesterol	2.217	−0.153–4.586	0.107	0.067
Low HDL-cholesterol	Reference			
Model 2				
High HDL-cholesterol	3.239	0.766–5.713	0.158	0.011
Medium HDL-cholesterol	2.162	−0.208–4.532	0.105	0.074
Low HDL-cholesterol	Reference			
APOE4-negative				
Model 1				
High HDL-cholesterol	2.230	−0.460–4.919	0.110	0.104
Medium HDL-cholesterol	1.161	−1.498–3.820	0.056	0.390
Low HDL-cholesterol	Reference			
Model 2				
High HDL-cholesterol	2.105	−0.640–4.849	0.104	0.132
Medium HDL-cholesterol	1.009	−1.675–3.693	0.049	0.459
Low HDL-cholesterol	Reference			
APOE4-positive				
Model 1				
High HDL-cholesterol	11.815	6.432–17.198	0.634	<0.001
Medium HDL-cholesterol	9.376	4.069–14.682	0.513	0.001
Low HDL-cholesterol	Reference			
Model 2				
High HDL-cholesterol	10.677	3.632–17.722	0.573	0.005
Medium HDL-cholesterol	8.576	2.235–14.917	0.470	0.010
Low HDL-cholesterol	Reference			

Abbreviations: B, regression coefficient, CI, confidence interval; β, standardized beta; *p*, *p*-value; APOE4, apolipoprotein E ε4 allele; EMS, episodic memory score; TS, total score of the Consortium to Establish a Registry for Alzheimer’s Disease; VRS, vascular risk score, BMI, body mass index; PASE, physical activity scale for the elderly. The first model included age, sex, APOE4, education, and clinical diagnosis as covariates; the second model included those covariates plus VRS, BMI, PASE total score, protein intake, fruit/vegetable, albumin, fasting glucose, and LDL-cholesterol.

**Table 6 nutrients-17-02321-t006:** Results of the multiple linear regression analyses of the association between the HDL-cholesterol level and cognitive decline according to APOE4 subgroup in older adults without a 3-month decline in food intake (*n* = 182).

	B	95% CI	β	*p*
TS				
Overall ^a^				
Model 1	0.145	0.015–0.274	0.125	0.029
Model 2	0.119	−0.012–0.250	0.103	0.074
APOE4-negative ^b^				
Model 1	−0.036	−0.173–0.102	−0.030	0.610
Model 2	−0.045	−0.183–0.093	−0.038	0.521
APOE4-positive ^b^				
Model 1	0.585	0.378–0.792	0.642	<0.001
Model 2	0.527	0.222–0.832	0.578	0.002
EMS				
Overall ^a^				
Model 1	0.116	0.038–0.194	0.166	0.004
Model 2	0.104	0.024–0.183	0.148	0.011
APOE4-negative ^b^				
Model 1	0.025	−0.059–0.109	0.037	0.556
Model 2	0.019	−0.067–0.104	0.027	0.665
APOE4-positive ^b^				
Model 1	0.379	0.222–0.535	0.576	<0.001
Model 2	0.354	0.130–0.579	0.539	0.004

Abbreviations: B, regression coefficient, CI, confidence interval; β, standardized beta; *p*, *p*-value; APOE4, apolipoprotein E ε4 allele; EMS, episodic memory score; TS, total score of the Consortium to Establish a Registry for Alzheimer’s Disease; VRS, vascular risk score, BMI, body mass index; PASE, physical activity scale for the elderly. ^a^ The first model included age, sex, APOE4, education, and clinical diagnosis as covariates; the second model included those covariates plus VRS, BMI, PASE total score, protein intake, fruit/vegetable, albumin, fasting glucose, and LDL-cholesterol. ^b^ The first model included age, sex, education, and clinical diagnosis as covariates; the second model included those covariates plus VRS, BMI, PASE total score, protein intake, fruit/vegetable, albumin, fasting glucose, and LDL-cholesterol.

## Data Availability

The study data are not freely accessible because the IRB of the Hallym University Dongtan Sacred Heart Hospital prevents public sharing of such data for privacy reasons. However, the data are available on reasonable request after IRB approval. Requests for data access can be submitted to an independent administrative coordinator by e-mail (yoon4645@gmail.com).

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
