# Peer review of "High-Density Lipoprotein Cholesterol and Cognitive Function in Older Korean Adults Without Dementia: Apolipoprotein E4 as a Moderating Factor"

_nutrients, 2025, doi:10.3390/nu17142321_

Round 1
Reviewer 1 Report
Comments and Suggestions for Authors
Certainly. Here is a reviewer’s report for the manuscript “High-Density Lipoprotein Cholesterol and Episodic Memory: Apolipoprotein E4 as a Moderating Factor” in the tone of Matthias B. Schulze—methodologically rigorous, cautiously interpretive, and committed to epidemiological clarity—while integrating perspectives from leading researchers in the field (e.g., Holtzman, Poirier, Pericak-Vance):
⸻
The manuscript is conceptually solid, methodologically detailed, and adds nuance to a field with heterogeneous findings. It strengthens the hypothesis that lipid metabolism may buffer or exacerbate genetic risk conferred by APOE4, building on prior work by Holtzman et al. and Poirier et al. However, the manuscript needs clarifications and reframing to improve scientific rigor, generalizability, and theoretical coherence.
Major points:
1. The cross-sectional nature limits causal inference. While the authors acknowledge this, action-oriented interpretations should be cautious. Language like “is associated with” or “suggests a potential mitigating association” is more appropriate.
2. The moderation effect of APOE4 is statistically robust, but the small APOE4+ group size (n=40) may limit power. Confidence intervals should be emphasized. Consider discussing possible Type I errors or false discovery rates given multiple comparisons.
3.While the manuscript covers various mechanistic pathways (e.g., ABCA1/ABCG1 transporters, PON1, S1P), this section feels encyclopedic. Prioritize pathways with strong human data, such as HDL-C’s modulation of amyloid clearance and vascular integrity (Mahley & Huang, 2012; Liu et al., 2013).e sThe manuscript should shift its focus from mechanistic assertion to hypothesis-generation.
Contextualize the findings with existing longitudinal and interventional trials, such as recent RCTs and Mendelian randomization studies, to address translational challenges.
Include variance inflation factors or collinearity diagnostics to reassure readers that multicollinearity among covariates did not bias estimates. Justify categorization of HDL-C levels using sex-specific cutoffs, especially considering the high female sample size. Conduct sensitivity analyses using continuous variables.
Contextualize Table S1 and S2 in the main text to ensure key insights are not relegated to the appendix.
Standardize terminology throughout and include a brief definition in the Methods section.
Discuss generalizability to the broader older adult population to address selection bias introduced by recruitment through memory clinics and community outreach.
Strengths:
- Rigorous control for vascular, nutritional, and lifestyle confounders.
- Use of validated cognitive assessments with high sensitivity to early cognitive decline.
- Integration of APOE4 genotyping allows for stratified analysis with clear clinical relevance.
Author Response
- The cross-sectional nature limits causal inference. While the authors acknowledge this, action-oriented interpretations should be cautious. Language like “is associated with” or “suggests a potential mitigating association” is more appropriate.
=> We thank the reviewer for this important comment. We fully agree that the cross-sectional design limits our ability to infer causality. In response, we have carefully revised the language throughout the manuscript to ensure our interpretations remain appropriately cautious (Abstract: p2, lines 39-45; Conclusions: p16, lines 422-427).
- The moderation effect of APOE4 is statistically robust, but the small APOE4+ group size (n=40) may limit power. Confidence intervals should be emphasized. Consider discussing possible Type I errors or false discovery rates given multiple comparisons.
=> Thank you for your insightful comment. We agree that the relatively small sample size in the APOE4-positive group (n = 40) may limit statistical power and affect the precision of the estimates. The wide confidence intervals around some of the beta coefficients reflect statistical uncertainty, and the results should therefore be interpreted with caution. Nevertheless, the consistency of findings across both Model 1 and Model 2—before and after adjusting for vascular, metabolic, and nutritional covariates—provides some support for the robustness of the observed associations. We have explicitly noted this limitation in the manuscript, along with the importance of interpreting results with attention to the wide confidence intervals (Discussion: p14, lines 403-409). Regarding multiple comparisons, we acknowledge the potential for type I error due to multiple comparisons. Although no formal correction was applied given the exploratory nature of the study, the association between HDL-C and episodic memory (EMS) remained statistically significant even after Bonferroni adjustment (p < 0.0125, based on four cognitive outcomes). This has been noted in the revised manuscript (Discussion: p14, lines 398-403).
3. While the manuscript covers various mechanistic pathways (e.g., ABCA1/ABCG1 transporters, PON1, S1P), this section feels encyclopedic. Prioritize pathways with strong human data, such as HDL-C’s modulation of amyloid clearance and vascular integrity (Mahley & Huang, 2012; Liu et al., 2013).e sThe manuscript should shift its focus from mechanistic assertion to hypothesis-generation.
=> Thank you for this insightful comment. We agree that prioritizing mechanisms with stronger translational relevance will improve the clarity and impact of our discussion. Accordingly, we have revised the paragraph to focus on human-supported mechanisms—particularly those involving amyloid clearance and vascular integrity—and reframed the mechanistic explanations to better support hypothesis-generation. Less directly substantiated pathways have been de-emphasized to avoid an encyclopedic tone (Discussion: p12-13, lines 330-354).
4. Contextualize the findings with existing longitudinal and interventional trials, such as recent RCTs and Mendelian randomization studies, to address translational challenges.
=> Thank you for this insightful comment. To improve translational relevance, we revised the Discussion to incorporate recent findings from longitudinal cohort studies, randomized controlled trials (RCTs), and Mendelian randomization (MR) studies examining HDL-C and cognitive outcomes. While cohort studies generally suggest a protective association, results from MR and RCTs have been inconsistent, reflecting the complexity of HDL-C functionality. These additions underscore the need for future studies that assess HDL quality as well as quantity and caution against overinterpretation of cross-sectional associations (Discussion: p13, lines 355–368; References 10, 53–55).
5. Include variance inflation factors or collinearity diagnostics to reassure readers that multicollinearity among covariates did not bias estimates.
=> We appreciate the reviewer’s concern regarding potential overfitting and multicollinearity. Covariates were selected based on prior evidence linking them to both HDL-C and cognitive function. Although Model 2 included a relatively large number of covariates, variance inflation factors (VIFs) for all variables were below 5, indicating no significant multicollinearity. This supports that the model was appropriately specified without collinearity concerns (Methods: p5, lines 197–200).
6. Justify categorization of HDL-C levels using sex-specific cutoffs, especially considering the high female sample size. Conduct sensitivity analyses using continuous variables.
=> Thank you for this important comment. HDL-C was categorized using sex-specific cutoffs based on Mayo Clinic guidelines to reflect clinically meaningful thresholds, particularly given that over 70% of our sample were women. This approach also ensured a balanced group distribution. To address concerns about generalizability and potential bias from categorization, HDL-C was analyzed as a continuous variable in our primary analyses. The findings were consistent across both approaches, supporting the robustness of the associations. This has been clarified in the manuscript (Discussion: p13-14, lines 386–393).
7. Contextualize Table S1 and S2 in the main text to ensure key insights are not relegated to the appendix.
=> Thank you for this valuable suggestion. To highlight key findings more clearly, we have incorporated the contents of Supplementary Tables S1 and S2 into the main text as Table 5 and Table 6, respectively. The numbering of subsequent tables has been updated accordingly to maintain consistency (Tables 5 and 6).
8. Standardize terminology throughout and include a brief definition in the Methods section.
=> Thank you for your helpful suggestion. We have standardized the terminology for cognitive outcomes (e.g., EMS, NMS, TS) throughout the manuscript. Brief definitions of these measures have also been added to the manuscript to improve clarity (Methods: p3, lines 122–133).
9. Discuss generalizability to the broader older adult population to address selection bias introduced by recruitment through memory clinics and community outreach.
=> Thank you for raising this important point. We acknowledge that recruitment through a memory clinic and community outreach may introduce selection bias, potentially limiting generalizability. Accordingly, we have added a statement in the Discussion to note this limitation and clarified that the findings may not fully represent the broader older adult population. We also emphasized the need for replication in population-based cohorts (Discussion: p14, lines 393-398).

Reviewer 2 Report
Comments and Suggestions for Authors
The study investigated the relationship between serum high-density lipoprotein cholesterol (HDL-C) levels and cognitive function, with a particular focus on episodic memory (EMS), non-memory scores (NMS), and total score (TS), assessing the moderating role of the APOE4 (apolipoprotein E ε4 allele) genotype. Higher HDL-C levels were positively associated with EMS and TS, especially in APOE4-positive individuals, suggesting a protective effect. The results presented in the manuscript highlights the specific effects of HDL-C on cognition, supported by its neuroprotective mechanisms including reduced inflammation, oxidative stress, and enhanced lipid metabolism. The results underscore the importance of APOE4 status in assessments of cognitive health related to lipid metabolism, and this represent the novelty of this study and for this reason I recommend publication. The manuscript is well written, methodology used are properly, the results are presented in a concise way and the conclusions supports the results.
Minor:
1). A sentence cannot begin with an abbreviation. Authors must read with attention their manuscript and revise these type of sentences ( R149; R284);
2). Table S1 are mentioned in the manuscript as supplementary file but on MDPI website the supplementary file missing. The authors must upload supplementary file;
3). The References does not written acoording to MDPI rules. Authors must write the References according to MDPI rules.
Author Response
The study investigated the relationship between serum high-density lipoprotein cholesterol (HDL-C) levels and cognitive function, with a particular focus on episodic memory (EMS), non-memory scores (NMS), and total score (TS), assessing the moderating role of the APOE4 (apolipoprotein E ε4 allele) genotype. Higher HDL-C levels were positively associated with EMS and TS, especially in APOE4-positive individuals, suggesting a protective effect. The results presented in the manuscript highlights the specific effects of HDL-C on cognition, supported by its neuroprotective mechanisms including reduced inflammation, oxidative stress, and enhanced lipid metabolism. The results underscore the importance of APOE4 status in assessments of cognitive health related to lipid metabolism, and this represent the novelty of this study and for this reason I recommend publication. The manuscript is well written, methodology used are properly, the results are presented in a concise way and the conclusions supports the results.
Minor:
1). A sentence cannot begin with an abbreviation. Authors must read with attention their manuscript and revise these type of sentences (R149; R284);
=> We appreciate the reviewer’s careful attention to language clarity. In response to the comment, we have revised all relevant sentences that began with abbreviations (e.g., HDL-C, APOE4, PCR) to ensure that no sentence starts with an abbreviation. Thank you for helping us improve the manuscript’s readability and presentation (Abstract: p2, line 35, 36, 39-42; Introduction: p2, lines 51, 68; Methods: p4, lines 164, 165, 166, 169, 172, 173; Discussions: p11, lines 298, 330, 334, 337; Conclusions: p14, lines 422-427).
2). Table S1 are mentioned in the manuscript as supplementary file but on MDPI website the supplementary file missing. The authors must upload supplementary file;
=> Thank you for your comment. The content of Table S1 has been incorporated into the main text as Table 5. Therefore, the supplementary file is no longer necessary (Table 5).
3). The References does not written acoording to MDPI rules. Authors must write the References according to MDPI rules.
=> Thank you for pointing this out. We have thoroughly revised the reference list to ensure consistency and adherence to the Nutrients journal style. Specifically, we corrected all journal names using the appropriate NLM abbreviations, standardized capitalization and punctuation, and reformatted all entries to comply fully with MDPI referencing guidelines, including the use of italics for journal titles and volumes, and boldface for publication years (References: 1-55).

Reviewer 3 Report
Comments and Suggestions for Authors
This is an interesting study on a topic of great actuality. It is done with an adequate number of individuals. May be these data will help to sort out people who should undergo an earlier and more intensive prevention.
Author Response
This is an interesting study on a topic of great actuality. It is done with an adequate number of individuals. May be these data will help to sort out people who should undergo an earlier and more intensive prevention.
=> We sincerely thank the reviewer for the positive and encouraging comments. We agree that our findings may contribute to identifying individuals at greater risk who may benefit from earlier and more intensive preventive strategies.

Reviewer 4 Report
Comments and Suggestions for Authors
The manuscript by Choe et al. explored the association between serum high-density lipoprotein cholesterol (HDL-C) levels and cognitive function, with a particular focus on the moderating role of Apolipoprotein E4 (APOE4) status. This is a timely study that raises important questions about HDL-C and cognitive domains in older adults, offering potentially useful insights. While the manuscript is informative and well-structured in parts, several aspects require substantial revision to improve clarity, enhance methodological transparency, and ensure that the results, discussion and conclusions are fully supported by the data. Below, I provide detailed comments organized by key sections of the manuscript.
1. Title
1) While the title emphasizes episodic memory, the Abstract presents findings related not only to episodic memory but also to global cognition, which was significantly associated with HDL-C levels and moderated by APOE4. This creates a degree of mismatch between the title and the aim of the study. To maintain consistency, the title could be expanded to include global cognition. Alternatively, the Abstract should clarify that episodic memory was the primary focus of the investigation, with global cognition analyzed as a secondary outcome
2) The current title does not indicate that the study is cross-sectional and observational, though including such details might make the title overly long. However, the study population, older Korean adults without dementia, is central to the research scope and interpretation. Including this in the title would help readers better understand the study’s context and applicability from the outset.
2. Abstract
1) As noted in the comment on the title, the Abstract should indicate that this was a cross-sectional study and clearly state the study population.
2) The Abstract does not mention whether the regression analyses accounted for potential confounding variables.
3. Introduction:
1) The role of APOE4 as a moderator is acknowledged in the Introduction. But this point could be more compellingly supported by citing specific evidence that demonstrates how few studies have explicitly tested moderation effects. The authors might consider explicitly framing this critical gap in the existing literature.
2) Although Alzheimer’s disease and cognitive decline are mentioned, the Introduction does not elaborate on the potential clinical relevance of understanding HDL-C and APOE4 interactions. Briefly discussing how this knowledge could inform risk stratification or early intervention strategies would enhance the rationale for the study.
3) The terms ‘cognitive function’ and ‘cognition’ are used interchangeably throughout the Introduction. While not incorrect, greater consistency would improve precision.
4) The aims of the study are somewhat embedded and could be more clearly framed.
4. Materials and Methods
1) Although the GLAD cohort is prospective, the present analysis is cross-sectional. This should be explicitly stated to avoid any misinterpretation.
2) Participants were recruited through clinical screening and community outreach, but the approach used to ensure representativeness is vaguely described as ‘...efforts were made...’ (Lines 85-86). This phrasing may confuse readers and should be clarified to describe the sampling strategy more specifically.
3) Information regarding the construct validity of the composite cognitive scores (EMS and NMS) is not presented in the text. While the scoring method appears reasonable, it would strengthen the methodology to either reference prior validation studies or provide a brief justification for the creation of these summary scores.
4) The criteria for categorizing dietary intake (e.g., protein, fruit or vegetables; Table 1) into ‘high’, ‘medium’, or ‘low’ are not defined. Please provide specific cutoffs or thresholds used for these classifications.
5) Lines 162–167: HDL-C was trichotomized into low (<50 mg/dL), medium (50–60 mg/dL), and high (>60 mg/dL) based on the Mayo Clinic criteria for women. The authors noted that this approach helped balance the distribution of participants across strata (L167). However, the Mayo Clinic criteria may not generalize well to men or to the Korean population. This limitation should be acknowledged, and the rationale for applying these specific thresholds should be more clearly justified. Please also refer to the relevant comments under the Results section for an additional important concern.
6) Lines 168–176:
(a) The regression models included a large number of covariates. Please briefly justify the rationale for their selection and inclusion in the models.
(b)APOE4-positive participants accounted for only 40 out of 196 total participants. Yet, Model 2 included a large number of covariates (approximately 14 variables) which raises concern, particularly for the stratified and interaction models, about possible overfitting and reduced statistical power. Although this reviewer is not a statistician, such modeling decisions warrant careful justification. Additionally, some covariates may be intercorrelated, yet there is no mention of checks for multicollinearity, such as variance inflation factors (VIFs).
7) Lines 180–184: The authors mentioned including APOE4 status as a covariate and also describe conducting separate analyses for APOE4-positive and APOE4-negative groups. However, it’s unclear how APOE4 was used in the models testing interaction effects—was it included as a main effect along with the interaction term? Please clarify this modeling approach, as a brief explanation would help this reviewer and potential readers understand the analysis more clearly.
8) The authors should confirm that they tested assumptions of linear regression.
9) It appears that HDL-C was treated both as a continuous and a categorical variable. Please clarify which analysis used which format.
5. Results
1) All beta coefficients reported in the tables are unstandardized. Including standardized coefficients would improve interpretability by allowing readers to compare the relative importance of predictors across models.
2) Lines 206 and 236: Table S1 is referenced in the text, but it is neither shown in the main manuscript nor included as a Supplement.
3) Lines 239–241: Table S2 is also referenced but is not provided or described, which limits the transparency and reproducibility of the sensitivity analysis.
4) Table 1:
(a) As previously noted, operational definitions for categorical dietary variables (e.g., ‘high’, ‘medium’, ‘low’ intake of protein and fruit/vegetables) are missing. These classifications should be clearly defined to allow meaningful interpretation.
(b) In Table 1, there appears to be an inconsistency in the reported sex distribution. Specifically, 50 out of 156 participants in the APOE4-negative group and 8 out of 40 in the APOE4-positive group are reported as female, totaling 58 females out of 196 participants. This corresponds to approximately 29.6%, not 70.4% as indicated under ‘Overall’. This discrepancy needs to be clarified, as it may have implications for subsequent analyses, particularly the HDL-C categorization, which is based on sex-specific thresholds.
(c) For the variable ‘Decrease in food intake over the past 3 months’, the reported percentages in the APOE4-negative group do not sum to 100%, suggesting a possible error or omission in reporting.
5) Table 2:
(a) HDL-C appears to be treated as a continuous variable in these models, but the unit of measurement and scaling are not clearly stated.
(b) Multiple cognitive outcomes were tested for association with HDL-C levels. Was any adjustment made for multiple comparisons (e.g., Bonferroni or false discovery rate correction)?
(c) The footnote lists abbreviations redundantly, repeating acronyms already introduced in the table.
5) Table 3:
The main effect of APOE4 appears to be large and negative in some models (e.g., for TS), yet this is not mentioned or discussed in the Results section. This is a potentially important finding, as it reinforces APOE4’s broad influence on cognitive performance and deserves explicit acknowledgment.
6) Table 4 and Figure 1:
(a) Presenting both Model 1 and Model 2 results for each APOE4 subgroup adds complexity without offering substantial interpretive value, as the differences between models are relatively minor. Additionally, the small sample size in the APOE4-positive group (n = 40) raises concerns about the stability and robustness of the estimates. Although some beta coefficients are large (e.g., B = 0.519), the wide confidence intervals further reflect statistical uncertainty.
(b) Table 4 and Figure 1 appear to present overlapping information regarding the association between HDL-C and cognitive outcomes stratified by APOE4 status. To improve clarity and avoid redundancy, the authors may consider consolidating these elements or enhancing cross-referencing between them to better integrate and streamline the presentation of results.
6. Discussion
1) Some of the language in the Discussion implies causality, which is not appropriate given the cross-sectional design of the study. For instance, Lines 266–267 state that “HDL-C might counteract APOE4’s detrimental effects…”, which suggests a causal relationship that cannot be supported by this study design.
2) The strong negative main effect of APOE4 on global cognition (e.g., Table 3: B = –22.733) is not addressed in the Discussion. This is a significant omission, as it may have important implications for interpreting the observed interaction effects and for understanding APOE4’s overall role in cognitive decline.
3) The small sample size of the APOE4-positive group (n = 40) is not acknowledged, despite this subgroup driving the most prominent findings.
4) The Discussion focuses almost exclusively on the positive findings related to episodic memory and global cognition. However, the non-significant results for other cognitive domains (e.g., verbal fluency, naming, and constructional praxis) are not addressed. Please provide a more balanced and comprehensive interpretation of the findings.
7. Conclusions (Lines 335–338)
The conclusions presented are too strong given the cross-sectional nature of the study. In particular, the suggestion for HDL-C-focused interventions feels premature, as no intervention was tested. Additionally, the study population, older Korean adults without dementia, may limit generalizability to broader or more diverse populations. The conclusion should be toned down to reflect the study’s observational design and avoid overstating the clinical implications.
8. References
In the references, there are inconsistencies in the use of full vs. abbreviated journal names, capitalization, and punctuation, all of which should be corrected for uniformity.
9. English and scientific writing
The manuscript is generally well-structured and readable, but several issues in verb tense, wordiness, and lack of specificity reduce clarity in parts of the text. While it’s not feasible to list all instances, the following are representative examples. An overall language review is recommended to enhance clarity and maintain consistency.
Lines 24–25: “We aim to investigate…” – Verb tense is inconsistent with the rest of the abstract, which is written in the past tense.
Lines 56–58: The phrase ‘rather than’ seems awkward and should be revised for clarity.
Line 70: The phrase ‘while examining the moderating role of APOE4 status’ is incomplete; moderating in what relationship?
Line 72: “...to mitigate genetic risks…”; ‘mitigate’ may not be the most appropriate term in this context.
Lines 91-96: “…one of four episodic memory subtests…” should be rephrased for clarity.
Line 230: The word ‘EMS’ is missing after ‘both’.
Author Response
The manuscript by Choe et al. explored the association between serum high-density lipoprotein cholesterol (HDL-C) levels and cognitive function, with a particular focus on the moderating role of Apolipoprotein E4 (APOE4) status. This is a timely study that raises important questions about HDL-C and cognitive domains in older adults, offering potentially useful insights. While the manuscript is informative and well-structured in parts, several aspects require substantial revision to improve clarity, enhance methodological transparency, and ensure that the results, discussion and conclusions are fully supported by the data. Below, I provide detailed comments organized by key sections of the manuscript.
- Title
1) While the title emphasizes episodic memory, the Abstract presents findings related not only to episodic memory but also to global cognition, which was significantly associated with HDL-C levels and moderated by APOE4. This creates a degree of mismatch between the title and the aim of the study. To maintain consistency, the title could be expanded to include global cognition. Alternatively, the Abstract should clarify that episodic memory was the primary focus of the investigation, with global cognition analyzed as a secondary outcome
=> Thank you for your insightful comment. To resolve the mismatch between the title and the abstract, we have revised the title to reflect the broader scope of cognitive outcomes, including both episodic memory and global cognition, which were significantly associated with HDL-C and moderated by APOE4. The new title ensures consistency with the study's aims and key findings (Title: p1, lines 1-2).
2) The current title does not indicate that the study is cross-sectional and observational, though including such details might make the title overly long. However, the study population, older Korean adults without dementia, is central to the research scope and interpretation. Including this in the title would help readers better understand the study’s context and applicability from the outset.
=> We also appreciate the suggestion to clarify the study population. Accordingly, we have updated the title to specify that the study was conducted in older Korean adults without dementia, which helps convey the research context and applicability more clearly to readers (Title: p1, lines 1-2).
- Abstract
1) As noted in the comment on the title, the Abstract should indicate that this was a cross-sectional study and clearly state the study population.
=> We appreciate the reviewer’s valuable suggestion. In response, we have revised the Abstract to specify that this was a cross-sectional study. Additionally, we now clearly state that the study population comprised dementia-free Korean older adults aged 65 to 90 years, recruited from both a memory clinic and community outreach efforts. These changes enhance clarity regarding the study design and sample characteristics (Abstract: p2, lines 26-28).
2) The Abstract does not mention whether the regression analyses accounted for potential confounding variables.
=> Thank you for this important point. We have revised the Methods section of the Abstract to explicitly state that multiple linear regression analyses were performed with adjustment for potential confounding variables, including age, sex, APOE4 status, education, diagnosis, vascular risk, nutritional status, physical activity, and blood biomarkers. This revision clarifies the rigor of our statistical approach (Abstract: p2, lines 30-33).
- Introduction:
1) The role of APOE4 as a moderator is acknowledged in the Introduction. But this point could be more compellingly supported by citing specific evidence that demonstrates how few studies have explicitly tested moderation effects. The authors might consider explicitly framing this critical gap in the existing literature.
=> Thank you for this valuable suggestion. We agree that highlighting the limited number of studies that have explicitly examined APOE4 as a moderator strengthens the rationale for our study. Accordingly, we have revised the Introduction to emphasize this gap and cited relevant literature to illustrate the scarcity of moderation analyses in prior research (Introduction: p3, lines 64-75).
2) Although Alzheimer’s disease and cognitive decline are mentioned, the Introduction does not elaborate on the potential clinical relevance of understanding HDL-C and APOE4 interactions. Briefly discussing how this knowledge could inform risk stratification or early intervention strategies would enhance the rationale for the study.
=> Thank you for this insightful comment. We agree that highlighting the clinical relevance of HDL-C and APOE4 interactions would strengthen the rationale for the study. Accordingly, we have revised the paragraph of the Introduction to briefly address how a better understanding of these interactions could support risk stratification and inform targeted early intervention strategies for individuals at heightened risk of cognitive decline or Alzheimer’s disease (Introduction: p3, lines 72-75).
3) The terms ‘cognitive function’ and ‘cognition’ are used interchangeably throughout the Introduction. While not incorrect, greater consistency would improve precision.
=> Thank you for pointing this out. We agree that consistent terminology improves clarity and precision. Accordingly, we have revised the Introduction to consistently use the term “cognitive function” throughout the section (Title page: p1, line 1; Abstract: p1, lines 40, 46; Introduction: p2, lines 70, 76, 77; Methods: p4, lines 176, 188, 189, 190; Results: p6, lines 220, 235, 258; Discussion: p11, lines 286, 298, 355, 356, 382; Conclusions: p14, line 422).
4) The aims of the study are somewhat embedded and could be more clearly framed.
=> Thank you for this helpful comment. To improve clarity, we have revised the final paragraph of the Introduction to state the study aims more explicitly and highlight the focus on domain-specific cognitive function and the moderating role of APOE4 (Introduction: p2, lines 76-79).
- Materials and Methods
1) Although the GLAD cohort is prospective, the present analysis is cross-sectional. This should be explicitly stated to avoid any misinterpretation.
=> Thank you for this important comment. We agree that clarifying the study design is essential to avoid misinterpretation. Accordingly, we have revised the Abstract, Methods (Study Design and Participants section), and Discussion to explicitly state that the present analysis is cross-sectional, based on baseline data from the prospective GLAD cohort (Methods: p2, lines 82, 83; Discussion: p13, lines 375-378).
2) Participants were recruited through clinical screening and community outreach, but the approach used to ensure representativeness is vaguely described as ‘...efforts were made...’ (Lines 85-86). This phrasing may confuse readers and should be clarified to describe the sampling strategy more specifically.
=> Thank you for this valuable suggestion. We have revised the description of our sampling strategy to clarify that participants were recruited through two parallel methods: (1) a structured dementia screening program at the memory clinic and (2) community outreach, including referrals by participants or their acquaintances. In the revised text, we specify that these community-based volunteers were carefully selected to reflect the demographic and cognitive diversity of the general older adult population, thereby enhancing the epidemiological representativeness of the sample (Methods: p2, lines 88-95).
3) Information regarding the construct validity of the composite cognitive scores (EMS and NMS) is not presented in the text. While the scoring method appears reasonable, it would strengthen the methodology to either reference prior validation studies or provide a brief justification for the creation of these summary scores.
=> Thank you for this helpful comment. We agree that providing justification for the use of composite cognitive scores would enhance the methodological rigor. We have revised the Methods section to briefly explain the rationale for calculating episodic memory score (EMS) and non-memory score (NMS), and we now reference prior studies that have employed similar composite scoring approaches using the CERAD battery in older Korean adults (References #1-#3).
References)
#1. Kim SG, Keum M, Choe YM, Suh GH, Lee BC, Kim HS, Lee JH, Hwang J, Yi D, Kim JW. Selenium and Episodic Memory: The Moderating Role of Apolipoprotein E ε4. Nutrients. 2025 Feb 6;17(3):595. doi: 10.3390/nu17030595. PMID: 39940451; PMCID: PMC11819958.
#2. Keum M, Lee BC, Choe YM, Suh GH, Kim SG, Kim HS, Hwang J, Yi D, Kim JW. Protein intake and episodic memory: the moderating role of the apolipoprotein E ε4 status. Alzheimers Res Ther. 2024 Aug 12;16(1):181. doi: 10.1186/s13195-024-01546-0. PMID: 39135146; PMCID: PMC11318328.
#3. Kim SG, Choe YM, Suh GH, Lee BC, Choi IG, Kim HS, Hwang J, Keum MS, Yi D, Kim JW. Manganese level and cognitive decline in older adults with the APOE e4 allele: a preliminary study. Psychiatry Res. 2023 Sep;327:115403. doi: 10.1016/j.psychres.2023.115403. Epub 2023 Aug 10. PMID: 37579537.
4) The criteria for categorizing dietary intake (e.g., protein, fruit or vegetables; Table 1) into ‘high’, ‘medium’, or ‘low’ are not defined. Please provide specific cutoffs or thresholds used for these classifications.
=> Thank you for your comment. We have added specific definitions for dietary intake classifications in the Methods section. Protein intake was categorized based on three MNA dietary markers, and fruit/vegetable intake was classified as ‘high’ if ≥2 servings/day were reported. These criteria are now clearly stated in the manuscript (Methods: p4, lines 150-155).
5) Lines 162–167: HDL-C was trichotomized into low (<50 mg/dL), medium (50–60 mg/dL), and high (>60 mg/dL) based on the Mayo Clinic criteria for women. The authors noted that this approach helped balance the distribution of participants across strata (L167). However, the Mayo Clinic criteria may not generalize well to men or to the Korean population. This limitation should be acknowledged, and the rationale for applying these specific thresholds should be more clearly justified. Please also refer to the relevant comments under the Results section for an additional important concern.
=> Thank you for this important comment. We agree that the HDL-C classification thresholds based on the Mayo Clinic criteria may not fully generalize to Korean men and women. Nevertheless, our primary analyses treated HDL-C as a continuous variable and yielded consistent associations with cognitive outcomes, supporting the robustness of our findings regardless of the categorization scheme. We selected these cutoffs to ensure sufficient participant distribution across groups, but we have now acknowledged this limitation in the revised Discussion section (Results: p9, lines 258-267; Discussion: p13-14, lines 386-393).
6) Lines 168–176:
(a) The regression models included a large number of covariates. Please briefly justify the rationale for their selection and inclusion in the models.
=> Thank you for this comment. We have clarified in the Methods section that the covariates included in the regression models were selected based on prior literature indicating their potential associations with both HDL-C levels and cognitive outcomes (e.g., age, APOE4, vascular risks, nutrition, etc.). Relevant references have also been added to support this rationale (Methods: p4, lines 188-193; References 13, 34-40).
(b)APOE4-positive participants accounted for only 40 out of 196 total participants. Yet, Model 2 included a large number of covariates (approximately 14 variables) which raises concern, particularly for the stratified and interaction models, about possible overfitting and reduced statistical power. Although this reviewer is not a statistician, such modeling decisions warrant careful justification. Additionally, some covariates may be intercorrelated, yet there is no mention of checks for multicollinearity, such as variance inflation factors (VIFs).
=> Thank you for your insightful comment. We agree that the relatively small sample size in the APOE4-positive group (n = 40) may limit statistical power and affect the precision of the estimates. The wide confidence intervals around some of the beta coefficients reflect statistical uncertainty, and the results should therefore be interpreted with caution. Nevertheless, the consistency of findings across both Model 1 and Model 2—before and after adjusting for vascular, metabolic, and nutritional covariates—provides some support for the robustness of the observed associations. We have explicitly noted this limitation in the manuscript, along with the importance of interpreting results with attention to the wide confidence intervals (Discussion: p14, lines 403-409).
We appreciate the reviewer’s concern regarding potential overfitting and multicollinearity. Covariates were selected based on prior evidence linking them to both HDL-C and cognitive function. Although Model 2 included a relatively large number of covariates, variance inflation factors (VIFs) for all variables were below 5, indicating no significant multicollinearity. This supports that the model was appropriately specified without collinearity concerns (Methods: p5, lines 197–200).
7) Lines 180–184: The authors mentioned including APOE4 status as a covariate and also describe conducting separate analyses for APOE4-positive and APOE4-negative groups. However, it’s unclear how APOE4 was used in the models testing interaction effects—was it included as a main effect along with the interaction term? Please clarify this modeling approach, as a brief explanation would help this reviewer and potential readers understand the analysis more clearly.
=> Thank you for raising this important point. In the models testing interaction effects, APOE4 was not treated as a covariate but rather as a moderator of the association between HDL-C and cognitive outcomes. Specifically, we included HDL-C, APOE4 status, and the HDL-C × APOE4 interaction term in the regression models, while adjusting for all other covariates. This approach allows for proper assessment of moderation effects. When a significant interaction was identified, we conducted stratified analyses within APOE4-positive and APOE4-negative groups to examine differential associations. We have revised the Methods section accordingly for clarity (Methods: p5, lines 201-207).
8) The authors should confirm that they tested assumptions of linear regression.
=> We thank the reviewer for this important comment. To ensure the robustness of our regression analyses, we confirmed that key assumptions of linear regression were met. We assessed the normality and homoscedasticity of residuals through normal probability plots and scatter plots, and verified linearity through visual inspection of residual trends. These diagnostic checks support the appropriateness of the linear regression models used in our analyses (Methods: p5, lines 197-200).
9) It appears that HDL-C was treated both as a continuous and a categorical variable. Please clarify which analysis used which format.
=> We appreciate the reviewer’s helpful observation. HDL-C was analyzed both as a continuous and a categorical variable, depending on the purpose of each analysis. In our primary regression analyses, HDL-C was treated as a continuous variable to examine its linear association with cognitive function. In additional analyses, HDL-C was categorized into three groups (‘low,’ ‘medium,’ and ‘high’) to facilitate interpretation in stratified comparisons and visual presentations. This dual approach has now been clarified in the manuscript (Methods: p4, lines 176-187).
- Results
1) All beta coefficients reported in the tables are unstandardized. Including standardized coefficients would improve interpretability by allowing readers to compare the relative importance of predictors across models.
=> Thank you for the valuable suggestion. As recommended, we have included standardized beta coefficients alongside the unstandardized coefficients in all relevant tables to improve interpretability and allow for comparison of the relative importance of predictors across models (Tables 2-6).
2) Lines 206 and 236: Table S1 is referenced in the text, but it is neither shown in the main manuscript nor included as a Supplement.
=> Thank you for pointing this out. The table previously referred to as Table S1 has now been included in the main manuscript as Table 5 to ensure clarity and accessibility. The corresponding references in the text have also been updated accordingly (Results: p9, line 258-267; Table 5).
3) Lines 239–241: Table S2 is also referenced but is not provided or described, which limits the transparency and reproducibility of the sensitivity analysis.
=> Thank you for pointing this out. The table previously referred to as Table S2 has now been included in the main manuscript as Table 6 to ensure clarity and accessibility. The corresponding references in the text have also been updated accordingly (Results: p10, line 270-273; Table 6).
4) Table 1:
(a) As previously noted, operational definitions for categorical dietary variables (e.g., ‘high’, ‘medium’, ‘low’ intake of protein and fruit/vegetables) are missing. These classifications should be clearly defined to allow meaningful interpretation.
=> Thank you for your comment. We have added specific definitions for dietary intake classifications in the Methods section. Protein intake was categorized based on three MNA dietary markers, and fruit/vegetable intake was classified as ‘high’ if ≥2 servings/day were reported. These criteria are now clearly stated in the manuscript (Methods: p4, lines 150-155).
(b) In Table 1, there appears to be an inconsistency in the reported sex distribution. Specifically, 50 out of 156 participants in the APOE4-negative group and 8 out of 40 in the APOE4-positive group are reported as female, totaling 58 females out of 196 participants. This corresponds to approximately 29.6%, not 70.4% as indicated under ‘Overall’. This discrepancy needs to be clarified, as it may have implications for subsequent analyses, particularly the HDL-C categorization, which is based on sex-specific thresholds.
=> Thank you for your careful review. We sincerely apologize for the error in the previous version of Table 1. Due to a typographical mistake, the number of male participants was mistakenly entered in the female column. This has now been corrected. The accurate sex distribution is 138 females out of 196 participants (70.41%), with 106 (67.95%) in the APOE4-negative group and 32 (80.00%) in the APOE4-positive group (Table 1).
(c) For the variable ‘Decrease in food intake over the past 3 months’, the reported percentages in the APOE4-negative group do not sum to 100%, suggesting a possible error or omission in reporting.
=> Thank you for your careful review. You are correct that the previously reported number for the “yes” response in the APOE4-negative group was erroneous. Specifically, it was mistakenly reported as 37 participants (23.72%) instead of the correct value of 11 participants (7.05%). We have corrected this error in the revised version of Table 1. The total percentage now appropriately sums to 100% for the APOE4-negative group (Table 1).
5) Table 2:
(a) HDL-C appears to be treated as a continuous variable in these models, but the unit of measurement and scaling are not clearly stated.
=> We appreciate the reviewer’s comment. To avoid confusion, we have now specified the unit of measurement (mg/dL) for HDL-C in the Methods section when treated as a continuous variable. In addition, the relevant tables where HDL-C was analyzed as a continuous variable have been clearly indicated in the text for clarity (Tables 2-4 and 6; Methods: p4, lines 176-197).
(b) Multiple cognitive outcomes were tested for association with HDL-C levels. Was any adjustment made for multiple comparisons (e.g., Bonferroni or false discovery rate correction)?
=> Thank you for this important comment. Regarding multiple comparisons, we acknowledge the potential for type I error due to multiple comparisons. Although no formal correction was applied given the exploratory nature of the study, the association between HDL-C and episodic memory (EMS) remained statistically significant even after Bonferroni adjustment (p < 0.0125, based on four cognitive outcomes). This has been noted in the revised manuscript (Discussion: p14, lines 398-403).
(c) The footnote lists abbreviations redundantly, repeating acronyms already introduced in the table.
=> Thank you for pointing this out. We have revised the footnote to eliminate redundant definitions of abbreviations that were already explained in the table. Only acronyms not introduced elsewhere in the table are now defined in the footnote for clarity and brevity (Table 2).
5) Table 3:
The main effect of APOE4 appears to be large and negative in some models (e.g., for TS), yet this is not mentioned or discussed in the Results section. This is a potentially important finding, as it reinforces APOE4’s broad influence on cognitive performance and deserves explicit acknowledgment.
=> Thank you for your comment. We have now explicitly described and discussed the significant main effects of APOE4 on global and episodic memory scores in both the Results and Discussion sections, as they underscore APOE4’s broad influence on cognitive performance (Results: p8, lines 242-246; Discussion: p16, lines 325-329).
6) Table 4 and Figure 1:
(a) Presenting both Model 1 and Model 2 results for each APOE4 subgroup adds complexity without offering substantial interpretive value, as the differences between models are relatively minor. Additionally, the small sample size in the APOE4-positive group (n = 40) raises concerns about the stability and robustness of the estimates. Although some beta coefficients are large (e.g., B = 0.519), the wide confidence intervals further reflect statistical uncertainty.
=> Thank you for your comment. Although significant associations were observed in the APOE4-positive subgroup, the relatively small sample size (n = 40) may limit the stability and generalizability of the estimates. The wide confidence intervals around some of the beta coefficients reflect statistical uncertainty, and the results should therefore be interpreted with caution. Nevertheless, the consistency of findings across both Model 1 and Model 2—before and after adjusting for vascular, metabolic, and nutritional covariates—provides some support for the robustness of the observed associations. This limitation has been addressed in the revised Discussion (Discussion: p14, lines 403-409).
(b) Table 4 and Figure 1 appear to present overlapping information regarding the association between HDL-C and cognitive outcomes stratified by APOE4 status. To improve clarity and avoid redundancy, the authors may consider consolidating these elements or enhancing cross-referencing between them to better integrate and streamline the presentation of results.
=> Thank you for your helpful suggestion. While Table 5 and Figure 1C–F present related results, they offer complementary perspectives—statistical estimates and visual illustrations. We have revised the Results section to enhance cross-referencing and clarify their respective roles in presenting the stratified associations (Results: p8, lines 253-255).
- Discussion
1) Some of the language in the Discussion implies causality, which is not appropriate given the cross-sectional design of the study. For instance, Lines 266–267 state that “HDL-C might counteract APOE4’s detrimental effects…”, which suggests a causal relationship that cannot be supported by this study design.
=> Thank you for the valuable comment. We have revised the wording to avoid causal implication by replacing “might counteract” with “may be associated with a reduced impact of,” reflecting the observational nature of the study (Discussion: p11, lines 298-301).
2) The strong negative main effect of APOE4 on global cognition (e.g., Table 3: B = –22.733) is not addressed in the Discussion. This is a significant omission, as it may have important implications for interpreting the observed interaction effects and for understanding APOE4’s overall role in cognitive decline.
=> We appreciate this important observation. In response, we have revised the Discussion to explicitly address the strong negative main effect of APOE4 on global cognition. This addition underscores APOE4’s established role in cognitive vulnerability and clarifies its relevance in interpreting the observed interaction effects with HDL-C (Discussions: p12, lines 325-329).
3) The small sample size of the APOE4-positive group (n = 40) is not acknowledged, despite this subgroup driving the most prominent findings.
=> We acknowledge the relatively small sample size of the APOE4-positive group (n = 40) and have added a statement in the Discussion to highlight this limitation. While the findings in this subgroup are noteworthy, we agree that caution is warranted when interpreting the results, and future studies with larger APOE4-positive samples are needed to confirm their robustness (Discussions: p14, lines 403-409).
4) The Discussion focuses almost exclusively on the positive findings related to episodic memory and global cognition. However, the non-significant results for other cognitive domains (e.g., verbal fluency, naming, and constructional praxis) are not addressed. Please provide a more balanced and comprehensive interpretation of the findings.
=> Thank you for your comment. We have revised the Discussion to include the non-significant results for other cognitive domains (e.g., verbal fluency, naming, and constructional praxis), highlighting the domain-specific nature of the observed associations and emphasizing the need for further investigation into these differential effects (Discussion: p12, lines 306-312).
- Conclusions (Lines 335–338)
The conclusions presented are too strong given the cross-sectional nature of the study. In particular, the suggestion for HDL-C-focused interventions feels premature, as no intervention was tested. Additionally, the study population, older Korean adults without dementia, may limit generalizability to broader or more diverse populations. The conclusion should be toned down to reflect the study’s observational design and avoid overstating the clinical implications.
=> Thank you for your valuable feedback. We have revised the conclusion to reflect the observational nature of the study, removed premature implications about interventions, and acknowledged the limited generalizability. The revised text now appropriately tones down the clinical implications (Abstract: p1, lines 39-45; Conclusion: p14, lines 422-427).
- References
In the references, there are inconsistencies in the use of full vs. abbreviated journal names, capitalization, and punctuation, all of which should be corrected for uniformity.
=> Thank you for pointing this out. We have thoroughly revised the reference list to ensure consistency and adherence to the Nutrients journal style. Specifically, we corrected all journal names using the appropriate NLM abbreviations, standardized capitalization and punctuation, and reformatted all entries to comply fully with MDPI referencing guidelines, including the use of italics for journal titles and volumes, and boldface for publication years (References: 1-55).
- English and scientific writing
The manuscript is generally well-structured and readable, but several issues in verb tense, wordiness, and lack of specificity reduce clarity in parts of the text. While it’s not feasible to list all instances, the following are representative examples. An overall language review is recommended to enhance clarity and maintain consistency.
=> Thank you for your constructive feedback. In response, we have thoroughly reviewed and revised the manuscript for clarity, conciseness, and consistency in language, with input from co-authors including native English speakers. The revised version reflects these improvements throughout the text.
Lines 24–25: “We aim to investigate…” – Verb tense is inconsistent with the rest of the abstract, which is written in the past tense.
=> Thank you for your comment. We have revised the verb tense for consistency, changing “We aim to investigate…” to the past tense to align with the rest of the abstract (Abstract: p1, line 25).
Lines 56–58: The phrase ‘rather than’ seems awkward and should be revised for clarity.
=> Thank you for your suggestion. To improve clarity and avoid potential misinterpretation, we have removed the phrase from the revised manuscript (Introduction: p2, line 61).
Line 70: The phrase ‘while examining the moderating role of APOE4 status’ is incomplete; moderating in what relationship?
=> Thank you for your comment. We have revised the sentence to clarify that APOE4 status was examined as a moderator in the relationship between serum HDL-C levels and cognitive function (Abstract: p1, line 39-42).
Line 72: “...to mitigate genetic risks…”; ‘mitigate’ may not be the most appropriate term in this context.
=> Thank you for your comment. We agree that "mitigate" may imply a direct effect on genetic risk, which is not appropriate in this context. We have revised the sentence to better reflect the potential protective role of HDL-C in relation to APOE4-associated cognitive vulnerability (Abstract: p1, line 42-44).
Lines 91-96: “…one of four episodic memory subtests…” should be rephrased for clarity.
=> Thank you for your helpful comment. We have revised the sentence to clarify that memory impairment was defined based on performance on one of four episodic memory subtests from the CERAD neuropsychological battery (Methods: p3, lines 100-104).
Line 230: The word ‘EMS’ is missing after ‘both’.
=> Thank you for pointing this out. We have corrected the sentence to include ‘EMS’ after ‘both’ for clarity (Results: p8, line 251).

Round 2
Reviewer 1 Report
Comments and Suggestions for Authors
No further comments
Author Response
No further comments.
=> We thank you for your time and positive evaluation.
Reviewer 4 Report
Comments and Suggestions for Authors
This reviewer would like to thank the authors for their thorough and thoughtful revisions to the manuscript. It is evident that the authors have carefully considered the feedback, and the improvements in clarity, structure, and methodological transparency are commendable.
As minor suggestions, it would be helpful to briefly acknowledge the gender imbalance in the study sample as a limitation, potentially alongside the sixth limitation already noted. Given the known sex differences in HDL metabolism and cognitive aging, this addition would provide important context for interpreting the findings.
Additionally, please explicitly state the threshold for statistical significance in the Statistical Analysis section. For clarity and completeness, it is also recommended to include a brief note regarding the use of Bonferroni adjustment for multiple comparisons.
Furthermore, please consider presenting the results in the same order as they appear in the corresponding table. For example, in Lines 221–224, EMS is discussed first, whereas it appears after TS in Table 2. Also, in the Results text, please clearly indicate when findings refer specifically to Model 2 (the fully adjusted model). Lastly, please ensure consistent footnotes across all tables, clearly defining statistical terms (e.g., B, β, p), model structures, and covariates included.
Author Response
This reviewer would like to thank the authors for their thorough and thoughtful revisions to the manuscript. It is evident that the authors have carefully considered the feedback, and the improvements in clarity, structure, and methodological transparency are commendable.
As minor suggestions, it would be helpful to briefly acknowledge the gender imbalance in the study sample as a limitation, potentially alongside the sixth limitation already noted. Given the known sex differences in HDL metabolism and cognitive aging, this addition would provide important context for interpreting the findings.
=> Thank you for your insightful comment. We addressed the issue of gender imbalance in the sixth limitation (Discussion: p14, lines 401-403).
Additionally, please explicitly state the threshold for statistical significance in the Statistical Analysis section. For clarity and completeness, it is also recommended to include a brief note regarding the use of Bonferroni adjustment for multiple comparisons.
=> Thank you for your comment. We have added a description of the statistical significance threshold in the Statistical Analysis section. In addition, a brief note on the use of Bonferroni correction (p < 0.0125 based on four cognitive outcomes) has been included for clarity, with detailed interpretation provided in the Discussion (Methods: p5, line 212-215).
Furthermore, please consider presenting the results in the same order as they appear in the corresponding table. For example, in Lines 221–224, EMS is discussed first, whereas it appears after TS in Table 2. Also, in the Results text, please clearly indicate when findings refer specifically to Model 2 (the fully adjusted model). Lastly, please ensure consistent footnotes across all tables, clearly defining statistical terms (e.g., B, β, p), model structures, and covariates included.
=> Thank you for your kind comment. We revised the Results text to present findings in the same order as they appear in the corresponding tables. Additionally, we have clearly indicated in the Results section when findings refer to Model 2 to enhance clarity. Lastly, we reviewed all tables to ensure consistency in the footnotes. Statistical terms such as B, β, and p are now clearly defined (Results: p6, lines 224-226; p8, lines 241, 242, 247, 248, 255-257; p9, 265, 269, 270; p10, lines 277; Discussion: p14, lines 405-409; Tables 1-6).